# Sadness, hopelessness and suicide attempts in bullying: Data from the 2018 Iowa youth survey

**Kaela L. Newman**[1], **Daniel S. Alexander**[1]**, John P. Rovers**[2]*

1 College of Arts and Sciences, Drake University, Des Moines, Iowa, United States of America, 2 College of Pharmacy & Health Sciences, Drake University, Des Moines, Iowa, United States of America

* John.Rovers@drake.edu

**Data Availability Statement:** The data underlying the results presented in the study are available from the Iowa Department of Public Health. Data for the IYS is collected by the State Departments of Education and Public Health. The data is cleaned and tabulated by the Iowa Consortium for

## Abstract

### Background

Bullying in schools is a common problem that can have significant consequences on the mental health of both bullies and victims of bullying. Some estimates suggest that 30% of American youth are bullied. Self-reported incidence of depression, anxiety, and suicide attempts has been correlated with bullying. Victims may also suffer from a variety of somatic complaints such as headache, sleep disorders, and others. Youth surveys undertaken by Education or Public Health Departments in most US states are an underutilized resource in evaluating the problem and any consequences.

### Objective

The objective of this study was to explore the association of being involved in bullying either as a victim and/or a bully on mental health and suicide ideation by analyzing data from the 2018 Iowa Youth Survey. The results will then be applied to the published anti-bullying literature to make suggestions for how anti-bullying programs may be designed.

### Methods

Data were obtained from the 2018 Iowa Youth Survey (IYS), which is a cross-sectional survey of 6th, 8th and 11th grade students. We chose two mental health questions as dependent variables and used a multivariable logistic regression analysis to evaluate the correlation between the two dependent variables and ten types of bullying included in the IYS. Since some respondents in the IYS were prescribed psychotropic medications to help with feeling angry, anxious, nervous, or sad, we adjusted for the use of psychotropic medication in our analysis. Similarly, the literature suggests that some students are both bullies and victims (bully-victims). Accordingly, we also adjusted for bully-victims.

### Results

Unadjusted Odds Ratios (ORs) showed that not all forms of bullying were correlated with a significant risk of mental distress. Physical bullying had comparatively little association

Substance Abuse Research and Evaluation at the University of Iowa, and the data remains the property of the Department of Public Health and considered confidential under State of Iowa, Department of Public Health Disclosure of Confidential Public Health Information, Records, or Data Policy#CO 01-16-002. The publicly available report is available from:https://iowayouthsurvey. idph.iowa.gov/Portals/20/IYS_Reports/1/ d4bcdac2-7e28-4c63-a5e6-06b48cd1a428.pdf.

**Funding:** The authors received no specific funding for this work.

**Competing interests:** The authors have declared that no competing interests exist.

(ORs < 1 or overlapped 1), while identity bullying on sexual orientation or gender identity or sexual joking was consistently correlated with significant ORs for feeling sad or hopeless and attempting suicide (ORs 1.40–2.84). Cyberbullying (ORs 1.32–1.70) and social bullying (ORs < 1–2.21) were correlated with mental distress with ORs generally between physical and identity bullying. When adjusting for medication use or bully-victim status, adjusted ORs (aORs) were generally lower than unadjusted ORs.

## Conclusions

Not all types of bullying were significantly correlated with feeling sad or hopeless or attempting suicide. Being able to evaluate the specific associations of different types of bullying may have implications for teachers or policy makers hoping to implement bullying mitigation strategies in their schools.

## Introduction

School bullying is a common problem and has been of longstanding interest to parents, teachers, school administrators, and students [1]. Estimates of the prevalence of bullying vary. The largest review of school bullying appears to be one by Koyanagi and colleagues [2]. In a sample of 134,229 students from 48 countries aged 12–15, they found the overall prevalence of bullying victimization and related suicide attempts and were 30.4% and 10.7% respectively. A review by Espelage and De La Rue found that the worldwide prevalence for bullying ranged from 10% to 27% and they estimated that 30% of American youth suffer from being bullied. A review by Dake and colleagues found that the rate of bullying was highest in elementary schools, lower in middle schools and lowest in high schools [3, 4]. Gan et al. found that 55% of students reported having been bullied [5]. A study in middle schoolers found that 82.7% of respondents felt that bullying was a problem and that 49.1% had been the victim of a bully [6]. A study of Indian school children suggested 31.4% of children experienced bullying [7].

Although the general public probably thinks it understands when bullying occurs, one definition in the literature defines bullying as, "A student is being bullied or victimized when he or she is exposed repeatedly and over time to negative actions on the part of one or more students" [3]. Others would add that there is a power differential between the bully and the bullied. Some authors, however, have noted that a power differential does not always exist and the definition that students have for bullying can vary substantially from the definition used by researchers in the field [3]. Pergolizzi and colleagues suggest that bullying may be separated into five domains: physical, verbal, social exclusion, spreading rumors and cyberbullying [6]. Since cyberbullying is often anonymous, victims do not necessarily know the identity of their bully. This suggests that the proposed requirement for a power differential may need to be revised.

Those involved in bullying can be separated into three subtypes. There are bullies, there are those who have been bullied (victims), and there are bullies who themselves have suffered from bullying (bully-victims) [3, 8, 9]. Such bully-victims have been found to have a higher risk of various somatic and psychological complaints [8].

Although historically, bullying was often treated as one of the unpleasant realities of growing up, recently, some authors have argued that bullying represents a global public health crisis because bullying can have serious consequences [10]. A recent WHO report substantiates this

concern [11]. Self-reported incidence of depression or anxiety has been found to be significantly correlated with having been bullied and symptoms worsen the more times a student suffers from being bullied [12]. A review by Rigby found that the consequences of being bullied result in a sense of low psychological well-being, poor social adjustment, psychological distress, and physical ill health [13].

A study in Arkansas found that bullying increased the risk of suicidal ideation and planning while a large meta-analysis reported that both traditional bullying and cyber-bullying were associated with an increased risk of deliberate self-harm [14, 15]. A study by Klomek et al. found that victims are at a higher risk for depression and suicidality and that bully-victims are at the highest risk [16].

According to the Centers for Disease Control and Prevention (CDC):

> We know that bullying behavior and suicide-related behavior are closely related. This means youth who report any involvement with bullying behavior are more likely to report high levels of suicide-related behavior than youth who do not report any involvement with bullying behavior. . . . It is correct to say that involvement in bullying, along with other risk factors, increases the chance that a young person will engage in suicide-related behaviors [17].

Other studies suggest that bullying victims suffer a wide variety of somatic complaints including headache, sleep disorders, bed wetting, fatigue and similar ailments [4, 7]. In addition, bullying victims have been found to use medication for pain and exhibit psychological problems more frequently than those who have not been bullied [18]. There is some evidence to suggest that children with attention-deficit/hyperactivity disorder (ADHD) may suffer from being bullied due to an inability to regulate their emotions, and that episodes of being bullied increase inversely with the ability to regulate emotions [19].

A potential resource to evaluate school yard bullying is the periodic statewide youth survey carried out by various state government departments in most US states [20–23]. In some states, the report may focus on crime and safety issues in youth [22] while other reports focus more on issues related to school, family, substance use and abuse, and other broad, social issues [20, 21, 23]. Often, the youth survey includes questions related to bullying, and if so, the data can provide an estimate for how large the problem is in a given state [20, 21].

Unfortunately, many youth surveys typically only report a tabulation of the results–e.g. in 2018, 31% of 6th graders, 27% of 8th graders, 19% of 10th graders and 17% of 12th graders reported being bullied [21]. Although this may give an estimate of the size of the problem, it does not provide any information on the consequences of bullying, the correlational factors, or the possible causes of bullying. In other words, although the results are tabulated, there may not be any actual analysis of the data. This can make it difficult for teachers and policy makers to create an effective means to mitigate bullying and its deleterious associations.

Although no one, single cause for bullying has been identified, the literature suggests a number of family, school, psychosocial, and other correlates [3, 4, 24, 25]. For example, if a child is brought up in a home with significantly disturbed family dynamics, and where problems are often solved with violence, such a child may be predisposed to see bullying as a means of protecting themselves and their interests [24].

Since the social, cultural, educational, and economic environment varies across all 50 US states, and since policies are often made at the state level, it then seems reasonable to undertake a more detailed analysis of bullying by means of a statewide youth survey. In this paper, we undertake such an analysis and use data from the 2018 Iowa Youth Survey to evaluate the relationship between bullying and feelings of sadness or hopelessness and suicide attempts [20].

## Objective

The objective of this study was to explore the correlations between involvement in bullying either as a victim and/or a bully and mental health and suicide ideation by analyzing data from the 2018 Iowa Youth Survey [20]. These results will then be applied to published studies of anti-bullying programs to illustrate how studies such as this one may have application for setting school anti-bullying policies.

## Methods

Data for this study were obtained from the 2018 Iowa Youth Survey (IYS) [20]. The IYS is a broad questionnaire offered every two or three years since 1999 to private and public-school students in the 6th, 8th, and 11th grade across the state of Iowa. It is a joint effort between the Iowa Department of Education (IDE) and the Iowa Department of Public Health (IDPH) as well as several other state bodies.

The 2018 survey has 212 questions over 7 sections and collects data on demographics, student experiences, alcohol use, tobacco use, marijuana use, gambling, mental health, student beliefs and attitudes, peers, the school environment, family setting, and the community.

No personal identifiers are collected. School district and county information are redacted by IDPH making personal identification of respondents impossible. In the 2018 study, a total of 74,830 records were received via a survey completed by students using Survey Monkey® online. The IYS assumes that each record represents one student. After data cleaning, 4,379 records were removed leaving 70,451 validated responses available for analysis. The next IYS took place in 2021 with results available too late for inclusion in this paper. Data cleaning and tabulation of the data is performed by the Iowa Consortium for Substance Abuse Research and Evaluation at the University of Iowa. The data remains the property of the Department of Public Health and considered confidential.

The IYS has 7 Yes/No questions that inquire into mental health (Table 1) and 10 questions that inquire into the experience of being bullied. (Table 2) We divided the 10 types of bullying into four categories: cyber, identity-based, physical, and social. (Table 2). Cyber bullying consists of bullying online or through social media. Bullying about one's race, religion, or sexual identity or gender orientation was considered to be identity bullying. Physical bullying involved physical violence, while social bullying was considered to be name calling, social exclusion or spreading falsehoods that did not take place online. This taxonomy differs from the 5 categories of bullying proposed by Pergolizzi but is more consistent with the questions asked in the IYS [6].

**Table 1. Mental health questions on IYS (n = 70,451).**

| Question Number | Question | Percent Answering Question | Percent Answering Yes |
|---|---|---|---|
| B60 | Has your doctor prescribed medicine for you because you feel angry, anxious, restless, nervous, or sad? | 97 | 16 |
| B61 | Do you currently take medicine as prescribed to help you not feel angry, anxious, restless, nervous, or sad? | 97 | 10 |
| B62 | During the past 12 months, did you ever feel so sad or hopeless every day for 2 weeks or more in a row that you stopped doing some usual activities? | 97 | 25 |
| B63 | During the past 12 months, have you thought about killing yourself? | 97 | 21 |
| B64 | During the past 12 months, have you made a plan about how you would kill yourself? | 97 | 10 |
| B65 | During the past 12 months, have you tried to kill yourself? | 97 | 5 |
| B66 | If you attempted to kill yourself during the past 12 months, did any attempt result in an injury or poisoning, or overdose that had to be treated by a doctor or a nurse? | 97 | 1 |

**Table 2. Types of bullying (n = 70,451).**

| Type of Bullying | Question # on IYS and Question | Percent Answering Question | Percent Reporting ≥ 1 Times |
|---|---|---|---|
| Cyber Bullying | E 20 I have received a threatening or hurtful message from another student in an email, on a website, on a cell phone, from text messaging, in an internet chat room, or in instant messaging | 95 | 9 |
| | E 21 Something hurtful has been shared about me on social media (Facebook, Twitter, Snapchat, etc.) | 95 | 8 |
| Identity Bullying | E 16 I was made fun of because of my race or color | 95 | 8 |
| | E17 I was made fun of because of my religion | 95 | 6 |
| | E18 I was made fun of because of my sexual orientation or gender identity | 95 | 6 |
| | E19 Other students made sexual jokes, comments, or gestures that hurt my feelings | 95 | 15 |
| Physical Bullying | E14 I was hit, kicked, pushed, shoved around, or locked indoors | 95 | 17 |
| Social Bullying | E12 I was called names, was made fun of, or teased in a hurtful way | 95 | 39 |
| | E13 Other students left me out of things on purpose, excluded me from their group of friends, or completely ignored me | 95 | 35 |
| | E 15 Other students told lies, spread false rumors about me, and tried to make others dislike me | 95 | 34 |

Of the 7 mental health questions, we chose 2 as dependent variables:

- During the past 12 months, did you ever feel so sad or hopeless almost every day for 2 weeks or more in a row that you stopped doing some usual activities? (Question B62)

- During the past 12 months, have you tried to kill yourself? (Question B65)

We chose these as the dependent variables for concision but also because sadness and hopelessness were thought to be the most common symptoms of mental illness in this population and are frequently related to depression [3, 8, 9]. Students who reported suicide attempts were assumed to have also thought about suicide and/or developed a suicide plan. All 10 bullying questions were included as independent variables.

Two questions (B60 and B61) inquired if a student was prescribed or was taking medicine to help them feel less angry, anxious, restless, nervous, or sad. Consequently, we controlled for psychotropic medication use. The IYS did not ask questions about specific medications prescribed or about the specific medical conditions.

Previous studies found that some bullies are also victims of bullying and refer to them as bully-victims, terminology we have adopted [3, 8, 9]. Thus, we also controlled for students who had engaged in bullying behavior and were themselves bullied.

As suggested by previous research, the consequences of being bullied may be more severe according to the number of experiences of being bullied a student has undergone [12]. The question stem for each bullying question asks how many times a respondent has experienced bullying and offers respondents a choice of six responses: 0 times, 1 time, 2 times, 3–5 times, 6–10 times, ≥ 11 times. In an attempt to evaluate if greater experience of bullying was correlated in our data, while not providing an overwhelming amount of data, we limited reporting each type of bullying to 0 times (referent), 1 time, 3–5 times and ≥ 11 times.

All variables were included in a multivariable logistic regression analysis to assess significant relationships between independent variables and the presence of mental health concerns. Descriptive statistics examined include 95% confidence intervals calculated for all demographic groups and bullying instances as well as an estimated odds ratio indicating the strength of association between independent variables and mental health concerns. We treated the

entire data set as a single cohort and did not attempt to undertake separate analyses for girls or boys, or grade level. Adjusted models included the following co-variables: being prescribed a psychotropic medication; taking a psychotropic medication; behaving as a bully. Correlations were considered significant if the 95% CI did not overlap 1.

## Ethical approval

This study (2020–21016) was reviewed and deemed exempt by the Drake University Institutional Review Board. The project was also approved by the IDPH.

## Results

A total of 70,451 validated responses were received to the most recent (2018) Iowa Youth Survey. The demographics of respondents are shown in Table 3.

## Data analysis

Unadjusted Odds Ratios (OR) for feelings of sadness or hopelessness and for suicide attempts after experiencing all 10 types of bullying are shown in Table 4. ORs and 95% Confidence Intervals are shown for a single experience, 3–5 experiences, and 11 or more experiences of being bullied. Adjusted ORs (aORs) are shown in Tables 5 (adjusted for psychotropic medication use) and 6 (adjusted for being a bully-victim).

Table 3. Demographics (n = 70,451).

| Demographic Data | Answered Yes or ≥ 1 time | % (n = 70,451) |
|---|---|---|
| Male | | 51% |
| Female | | 49% |
| 6th grade | | 36% |
| 8th grade | | 35% |
| 11th grade | | 30% |
| Race | | 5% |
| • White | | 76% |
| • Black/African American | | 5% |
| • American Indian or Alaskan Native | | 1% |
| • Asian | | 3% |
| • Native Hawaiian or Other Pacific Islander | | 0% |
| • Mixed or Multiple Race | | 7% |
| • Other Race | | 7% |
| Hispanic or Latino | | 13% |
| Has your doctor prescribed medicine for you because you feel angry, anxious, restless, nervous, or sad? YES | Yes | 16% |
| Do you currently take medicine as prescribed to help you not feel angry, anxious, restless, nervous, or sad? YES | Yes | 10% |
| During the past 12 months, did you ever feel so sad or hopeless every day for 2 weeks or more in a row that you stopped doing some usual activities? YES | Yes | 25% |
| During the past 12 months, have tried to kill yourself? YES | Yes | 5% |
| In the last 30 days, how many times have you bullied someone else at school? | ≥ 1 time | 12% |

**Table 4. Unadjusted odds ratios of undergoing type of bullying and feelings of sadness or hopelessness and suicide attempt (Referent is 0 in each case).**

| In the last 30 days, how many times have you been bullied at school in the ways listed below | During the past 12 months, did you ever feel so sad or hopeless every day for 2 weeks or more in a row that you stopped doing some usual activities? | | During the past 12 months, have tried to kill yourself? | |
|---|---|---|---|---|
| | Odds Ratio (95% CI) | P value | Odds Ratio (95% CI) | P value |
| E12 I was called names, was made fun of or teased in a hurtful way | | | | |
| 1 time | 1.23 (1.16–1.31) | <0.0001 | 1.06 (0.92–1.22) | 0.3862 |
| 3–5 times | 1.40 (1.29–1.52) | <0.0001 | 1.36 (1.16–1.59) | <0.0001 |
| ≥ 11 times | 1.35 (1.22–1.49) | <0.0001 | 1.37 (1.15–1.64) | 0.0004 |
| E13 Other students left me out of things on purpose, excluded me from their group of friends, or completely ignored me | | | | |
| 1 time | 1.26 (1.18–1.33) | <0.0001 | 0.84 (0.73–0.96) | 0.0139 |
| 3–5 times | 1.72 (1.58–1.86) | <0.0001 | 1.14 (0.97–1.33) | 0.092 |
| ≥ 11 times | 2.21 (1.98–2.46) | <0.0001 | 1.42 (1.19–1.70) | <0.0001 |
| E14 I was hit, kicked, pushed, shoved around, or locked indoors | | | | |
| 1 time | 0.92 (0.85–0.99) | 0.0368 | 1.15 (1.00–1.32) | 0.0353 |
| 3–5 times | 0.86 (0.76–0.973) | 0.0161 | 1.12 (0.93–1.34) | 0.22 |
| ≥ 11 times | 0.84 (0.73–0.98) | 0.0318 | 1.21 (0.98–1.50) | 0.0714 |
| E 15 Other students told lies, spread false rumors about me, and tried to make others dislike me | | | | |
| 1 time | 1.41 (1.33–1.49) | <0.0001 | 1.37 (1.20–1.56) | <0.0001 |
| 3–5 times | 1.70 (1.56–1.85) | <0.0001 | 1.91 (1.63–2.24) | <0.0001 |
| ≥ 11 times | 2.00 (1.79–2.24) | <0.0001 | 2.15 (1.79–2.58) | <0.0001 |
| E 16 I was made fun of because of my race or color | | | | |
| 1 time | 1.23 (1.11–1.36) | <0.0001 | 1.10 (0.92–1.30) | 0.2637 |
| 3–5 times | 1.33 (1.11–1.59) | 0.0013 | 1.34 (1.04–1.72) | 0.0201 |
| ≥ 11 times | 0.97 (0.81–1.17) | 0.8047 | 1.13 (0.87–1.46) | 0.337 |
| E17 I was made fun of because of my religion | | | | |
| 1 time | 0.99 (0.89–1.11) | 0.9833 | 0.90 (0.74–1.10) | 0.328 |
| 3–5 times | 0.92 (0.73–1.15) | 0.4912 | 1.01 (0.74–1.36) | 0.9394 |
| ≥ 11 times | 0.49 (0.38–0.63) | <0.0001 | 0.51 (0.37–0.71) | <0.0001 |
| E18 I was made fun of because of my sexual orientation or gender identity | | | | |
| 1 time | 2.10 (1.85–2.37) | <0.0001 | 2.25 (1.87–2.70) | <0.0001 |
| 3–5 times | 2.57 (2.07–3.19) | <0.0001 | 2.23 (1.71–2.88) | <0.0001 |
| ≥ 11 times | 2.05 (1.68–2.50) | <0.0001 | 2.84 (2.26–3.57) | <0.0001 |
| E19 Other students made sexual jokes, comments, or gestures that hurt my feelings | | | | |
| 1 time | 1.40 (1.30–1.51) | <0.0001 | 1.44 (1.26–1.65) | <0.0001 |
| 3–5 times | 1.60 (1.41–1.82) | <0.0001 | 1.63 (1.35–1.95) | <0.0001 |
| ≥ 11 times | 1.52 (1.31–1.77) | <0.0001 | 1.54 (1.25–1.88) | <0.0001 |
| E 20 I have received a threatening or hurtful message from another student in an email, on a website, on a cell phone, from text messaging, in an internet chat room, or in instant messaging | | | | |
| 1 time | 1.32 (1.20–1.45) | <0.0001 | 1.35 (1.16–1.58) | 0.0001 |
| 3–5 times | 1.52 (1.26–1.82) | <0.0001 | 1.70 (1.34–2.14) | <0.0001 |
| ≥ 11 times | 1.27 (1.02–1.58) | 0.0285 | 1.61 (1.24–2.08) | 0.0003 |
| E 21 Something hurtful has been shared about me on social media (Facebook, Twitter, Snapchat, etc.) | | | | |
| 1 time | 1.24 (1.13–1.36) | <0.0001 | 1.28 (1.10–1.50) | 0.0012 |

(*Continued*)

**Table 4.** (Continued)

| In the last 30 days, how many times have you been bullied at school in the ways listed below | During the past 12 months, did you ever feel so sad or hopeless every day for 2 weeks or more in a row that you stopped doing some usual activities? | | During the past 12 months, have tried to kill yourself? | |
|---|---|---|---|---|
| | Odds Ratio (95% CI) | P value | Odds Ratio (95% CI) | P value |
| 3–5 times | 1.19 (0.98–1.45) | 0.0752 | 1.27 (0.98–1.64) | 0.0627 |
| ≥ 11 times | 1.22 (0.97–1.54) | 0.0764 | 1.88 (1.45–2.45) | <0.0001 |

95% Odds ratios that overlap 1.0 are not considered significant

## Unadjusted odds ratios

**Physical bullying.** Unadjusted ORs indicated that students who reported being physically bullied were no more likely to reports feelings of sadness or hopelessness than students who reported no instances of being bullied.

A single experience of physical bullying resulted in an OR of 1.15 for attempting suicide. Although the ORs for suicide attempts increased with greater experiences of being bullied, the ORs overlapped 1 and therefore are not considered significant.

**Social bullying.** Social bullying resulted in ORs for feeling sad or hopeless of 1.23 to 2.21. ORs for social exclusion and false rumors increased with the greater number of times students reported being bullied and such students were twice as likely to report feeling sad or hopeless.

The associations of social bullying on suicide attempts generally increased with greater experiences of being bullied. ORs for name calling and social exclusion were significant at 1.36 to 1.42 but significance was only reached after 3 to 5 or ≥11 experiences of being bullied respectively. Lies and false rumors appeared to have the biggest association on suicide attempts with ORs of 1.37 to 2.15. Even a single episode of rumor mongering significantly increased the odds of a student attempting suicide.

**Identity bullying.** The associations of identity bullying varied depending on what aspect of a student's identity was targeted. Making fun of a student's religion indicated no significant association on feeling sad or hopeless.

Racial bullying increased feelings of sadness and hopelessness for single and 3–5 bullying episodes, but those racially bullied 11 or more times appeared no more likely to feel sad or hopeless than students who had not been racially bullied.

Bullying related to sexual orientation or gender identity or hurtful sexual comments had the biggest association of any type of bullying, identity or otherwise on feeling sad or hopeless. Those bullied for their sexual orientation or gender identity had ORs of 2.05 to 2.57. ORs for hurtful sexual comments were lower but were still between 1.40 to 1.60.

The associations of identity bullying on suicide attempts were also variable. Those bullied for their religion had ORs that differed little from those not bullied at all. Students racially bullied 3–5 times had ORs of 1.34 while the ORs for a single experience or ≥ 11 experiences had ORs that overlapped 1 and were not considered significant.

All forms and experiences of sexual or gender-based bullying were significant for suicide attempts and had ORs ranging from 1.44 to 2.84.

**Cyberbullying.** Students who reported cyberbullying were 1.22 to 1.52 times more likely to report feeling sad or hopeless than students who had not undergone such bullying. For both

**Table 5. Adjusted odds ratios of undergoing type of bullying and feelings of sadness or hopelessness and suicide attempt adjusted for psychotropic use.** (Referent is 0 in each case).

| In the last 30 days, how many times have you been bullied at school in the ways listed below | During the past 12 months, did you ever feel so sad or hopeless every day for 2 weeks or more in a row that you stopped doing some usual activities? | | | | During the past 12 months, have you tried to kill yourself? | | | |
|---|---|---|---|---|---|---|---|---|
| | Has your doctor prescribed medicine for you because you feel angry, anxious, restless, nervous, or sad? | | Do you currently take medicine as prescribed to help you not feel angry, anxious, restless, nervous, or sad? | | Has your doctor prescribed medicine for you because you feel angry, anxious, restless, nervous, or sad? | | Do you currently take medicine as prescribed to help you not feel angry, anxious, restless, nervous, or sad? | |
| | Adjusted Odds Ratio (95% CI) | | | | Adjusted Odds Ratio (95% CI) | | | |
| | Adjusted Odds Ratio (95% CI) | P value | Adjusted Odds Ratio (95% CI) | P value | Adjusted Odds Ratio (95% CI) | P value | Adjusted Odds Ratio (95% CI) | P value |
| **E12 I was called names, was made fun of, or teased in a hurtful way** | | | | | | | | |
| 1 time | 0.96 (0.84–1.10) | 0.6092 | 0.90 (0.76–1.07) | 0.2435 | 0.83 (0.67–1.02) | 0.0956 | 0.79 (0.61–1.02) | 0.0816 |
| 3–5 times | 1.07 (0.90–1.27) | 0.3971 | 1.09 (0.88–1.33) | 0.3990 | 1.10 (0.88–1.38) | 0.3610 | 1.03 (0.79–1.35) | 0.7900 |
| ≥ 11 times | 0.88 (0.71–1.07) | 0.2225 | 0.94 (0.73–1.21) | 0.6412 | 0.96 (0.75–1.24) | 0.7934 | 0.92 (0.68–1.25) | 0.6167 |
| **E13 Other students left me out of things on purpose, excluded me from their group of friends, or completely ignored me** | | | | | | | | |
| 1 time | 0.97 (0.84–1.11) | 0.6706 | 1.03 (0.87–1.22) | 0.7168 | 0.84 (0.69–1.03) | 0.1076 | 0.81 (0.63–1.03) | 0.09578 |
| 3–5 times | 1.32 (1.13–1.60) | 0.0006 | 1.41 (1.14–1.74) | 0.0012 | 1.04 (0.83–1.29) | 0.7229 | 1.07 (0.82–1.38) | 0.7909 |
| ≥ 11 times | 1.81 (1.45–2.26) | <0.0001 | 1.80 (1.37–2.36) | <0.0001 | 1.16 (0.90–1.50) | 0.2220 | 1.22 (0.90–1.65) | 0.1836 |
| **E14 I was hit, kicked, pushed, shoved around, or locked indoors** | | | | | | | | |
| 1 time | 0.80 (0.69–0.94) | 0.0077 | 0.79 (0.65–0.97) | 0.0244 | 1.02 (0.83–1.24) | 0.8431 | 0.99 (0.77–1.25) | 0.9353 |
| 3–5 times | 0.67 (0.53–0.85) | 0.0010 | 0.52 (0.38–0.70) | <0.0001 | 1.01 (0.77–1.32) | 0.9164 | 0.82 (0.58–1.14) | 0.26248 |
| ≥ 11 times | 0.72 (0.54–0.96) | 0.0282 | 0.65 (0.46–0.92) | 0.0165 | 1.00 (0.73–1.36) | 0.9684 | 0.82 (0.56–1.18) | 0.2958 |
| **E 15 Other students told lies, spread false rumors about me, and tried to make others dislike me** | | | | | | | | |
| 1 time | 1.41 (1.23–1.61) | <0.0001 | 1.41 (1.20–1.66) | <0.0001 | 1.07 (0.87–1.29) | 0.4970 | 1.18 (0.94–1.49) | 0.1425 |
| 3–5 times | 1.76 (1.48–2.11) | <0.0001 | 1.63 (1.30–2.03) | <0.0001 | 1.44 (1.15–1.80) | 0.0012 | 1.61 (1.23–2.09) | 0.0004 |
| ≥ 11 times | 2.05 (1.65–2.56) | <0.0001 | 1.69 (1.30–2.20) | <0.0001 | 1.52 (1.19–1.96) | 0.009 | 1.77 (1.31–2.38) | 0.0002 |
| **E 16 I was made fun of because of my race or color** | | | | | | | | |
| 1 time | 1.10 (0.88–1.38) | 0.3797 | 1.00 (0.75–1.33) | 0.9773 | 1.44 (1.12–1.84) | 0.0034 | 1.54 (1.13–2.07) | 0.0047 |
| 3–5 times | 1.22 (0.85–1.78) | 0.2688 | 1.18 (0.76–1.87) | 0.4456 | 1.26 (0.87–1.81) | 0.2036 | 1.45 (0.93–2.23) | 0.0864 |
| ≥ 11 times | 0.68 (0.48–0.97) | 0.0339 | 0.60 (0.39–0.93) | 0.0225 | 1.45 (1.00–2.09) | 0.0434 | 1.30 (0.83–2.02) | 0.2356 |
| **E17 I was made fun of because of my religion** | | | | | | | | |
| 1 time | 0.95 (0.75–1.22) | 0.7313 | 1.01 (0.75–1.36) | 0.9268 | 0.87 (0.64–1.16) | 0.3585 | 0.82 (0.57–1.55) | 0.2698 |

*(Continued)*

**Table 5.** (Continued)

| | During the past 12 months, did you ever feel so sad or hopeless every day for 2 weeks or more in a row that you stopped doing some usual activities? | | | | During the past 12 months, have you tried to kill yourself? | | | |
|---|---|---|---|---|---|---|---|---|
| **3–5 times** | 0.83 (90.55–1.27) | 0.3940 | 0.88 (0.52–1.50) | 0.6382 | 1.14 (0.75–1.72) | 0.5171 | 0.93 (0.54–1.55) | 0.7855 |
| **≥ 11 times** | 0.63 (0.41–0.97) | 0.0390 | 0.63 (0.37–1.07) | 0.0878 | 0.44 (0.28–0.70) | 0.00056 | 0.53 (0.31–0.90) | 0.0218 |
| **E18 I was made fun of because of my sexual orientation or gender identity** | | | | | | | | |
| **1 time** | 1.65 (1.30–2.10) | <0.0001 | 1.67 (1.28–2.24) | 0.0004 | 1.50 (1.14–1.94) | 0.0025 | 1.41 (1.02–1.92) | 0.0319 |
| **3–5 times** | 2.49 (1.70–3.75) | <0.0001 | 2.51 (1.58–4.16) | 0.0001 | 1.87 (1.32–2.63) | 0.004 | 1.93 (1.26–2.90) | 0.0019 |
| **≥ 11 times** | 2.26 (1.61–3.23) | <0.0001 | 2.99 (1.96–4.68) | <0.0001 | 2.11 (1.55–2.86) | <0.0001 | 1.85 (1.27–2.67) | 0.0010 |
| **E19 Other students made sexual jokes, comments, or gestures that hurt my feelings** | | | | | | | | |
| **1 time** | 1.28 (1.09–1.51) | 0.0018 | 1.36 (1.12–1.68) | 0.0018 | 1.29 (1.06–1.57) | 0.0100 | 1.33 (1.04–1.68) | 0.0173 |
| **3–5 times** | 1.25 (1.00–1.59) | 0.0543 | 1.23 (0.94–1.63) | 0.1311 | 1.58 (1.23–2.02) | 0.0003 | 1.62 (1.20–2.16) | 0.0011 |
| **≥ 11 times** | 1.39 (1.06–1.84) | 0.0176 | 1.41 (1.02–1.96) | 0.0378 | 1.42 (1.07–1.87) | 0.0130 | 1.68 (1.21–2.33) | 0.0018 |
| **E 20 I have received a threatening or hurtful message from another student in an email, on a website, on a cell phone, from text messaging, in an internet chat room, or in instant messaging** | | | | | | | | |
| **1 time** | 1.35 (1.11–1.65) | 0.0029 | 1.27 (0.99–1.69) | 0.0555 | 1.35 (1.08–1.69) | 0.0078 | 1.27 (0.96–1.67) | 0.0893 |
| **3–5 times** | 1.31 (0.95–1.83) | 0.0992 | 1.40 (0.95–2.09) | 0.0933 | 1.43 (1.03–1.96) | 0.0274 | 1.75 (1.20–2.53) | 0.0032 |
| **≥ 11 times** | 1.26 (0.85–1.89) | 0.2519 | 1.37 (0.86–2.21) | 0.1804 | 1.99 (1.38–2.86) | 0.0002 | 1.82 (1.19–2.78) | 0.0049 |
| **E 21 Something hurtful has been shared about me on social media (Facebook–Twitter–Snapchat–etc.)** | | | | | | | | |
| **1 time** | 1.46 (1.20–1.77) | 0.0001 | 1.67 (1.31–2.14) | <0.0001 | 1.19 (0.95–1.47) | 0.2677 | 1.07 (0.82–1.40) | 0.5760 |
| **3–5 times** | 1.44 (0.99–2.11) | 0.0558 | 1.44 (0.93–2.26) | 0.1008 | 1.37 (0.97–1.94) | 0.0693 | 1.30 (0.85–1.95) | 0.2104 |
| **≥ 11 times** | 1.27 (0.86–1.90) | 0.2262 | 1.25 (0.79–1.98) | 0.3327 | 1.99 (1.39–2.85) | 0.00012 | 2.03 (1.33–3.06) | 0.0009 |

95% Odds ratios that overlap 1 are not considered significant

types of cyberbullying, a single episode was enough to result in sadness or hopelessness but the ORs did not always significantly increase with greater experience of being bullied.

The associations of cyberbullying on suicide attempts were similar with ORs of 1.28 to 1.88 depending on the type and number of bullying experiences.

## Adjusted odds ratios

**Psychotropic medication use.** The IYS asks if students have been prescribed or are taking medications to help with feelings of anger, anxiety, restlessness, nervousness, or sadness. Since

**Table 6. Adjusted odds ratios of feelings of sadness or hopelessness and suicide attempt adjusted for bully-victims (Referent is 0 in each case).**

| In the last 30 days, how many times have you been bullied at school in the ways listed below | During the past 12 months, did you ever feel so sad or hopeless every day for 2 weeks or more in a row that you stopped doing some usual activities? | | During the past 12 months, have tried to kill yourself? | |
|---|---|---|---|---|
| | Odds Ratio (95% CI) | P value | Odds Ratio (95% CI) | P value |
| **E12 I was called names, was made fun of or teased in a hurtful way** | | | | |
| 1 time | 0.83 (0.71–0.97) | 0.0255 | 0.72 (0.54–0.95) | 0.0241 |
| 3–5 times | 1.05 (0.87–1.27) | 0.5519 | 0.85 (0.63–1.15) | 0.3033 |
| ≥ 11 times | 0.96 (0.77–1.18) | 0.7108 | 0.92 (0.67–1.27) | 0.6515 |
| **E13 Other students left me out of things on purpose, excluded me from their group of friends, or completely ignored me** | | | | |
| 1 time | 1.24 (1.06–1.44) | 0.0066 | 0.62 (0.46–0.82) | 0.0011 |
| 3–5 times | 1.45 (1.20–1.76) | 0.0001 | 0.75 (0.55–1.02) | 0.0709 |
| ≥ 11 times | 1.95 (1.53–2.44) | <0.0001 | 1.12 (0.80–1.55) | 0.4827 |
| **E14 I was hit, kicked, pushed, shoved around, or locked indoors** | | | | |
| 1 time | 0.82 (0.70–0.96) | 0.0141 | 1.36 (1.06–1.73) | 0.01133 |
| 3–5 times | 0.77 (0.61–0.96) | 0.02230 | 1.19 (0.86–1.62) | 0.2841 |
| ≥ 11 times | 0.77 (0.59–1.01) | 0.0666 | 1.08 (0.75–1.54) | 0.64689 |
| **E 15 Other students told lies, spread false rumors about me, and tried to make others dislike me** | | | | |
| 1 time | 1.29 (1.11–1.50) | 0.0009 | 1.16 (0.88–1.50) | 0.2678 |
| 3–5 times | 1.55 (1.28–1.88) | <0.0001 | 1.28 (0.93–1.74) | 0.1167 |
| ≥ 11 times | 2.27 (1.80–2.87) | <0.0001 | 1.36 (0.97–1.92) | 0.0722 |
| **E 16 I was made fun of because of my race or color** | | | | |
| 1 time | 1.18 (0.95–1.45) | 0.1169 | 1.03 (0.75–1.39) | 0.8503 |
| 3–5 times | 1.33 (0.96–1.84) | 0.0790 | 1.85 (1.25–2.69) | 0.0014 |
| ≥ 11 times | 1.01 (0.73–1.40) | 0.9088 | 0.87 (0.56–1.33) | 0.5476 |
| **E17 I was made fun of because of my religion** | | | | |
| 1 time | 1.26 (1.00–1.60) | 0.0471 | 1.08 (0.76–1.50) | 0.6344 |
| 3–5 times | 0.97 (0.61–1.54) | 0.9155 | 1.31 (0.77–2.15) | 0.2975 |
| ≥ 11 times | 0.56 (0.36–0.87) | 0.0102 | 0.60 (0.35–1.00) | 0.0572 |
| **E18 I was made fun of because of my sexual orientation or gender identity** | | | | |
| 1 time | 1.59 (1.22–2.08) | 0.0006 | 1.86 (1.31–2.62) | 0.0004 |
| 3–5 times | 1.41 (0.96–2.10) | 0.0816 | 1.76 (1.10–2.74) | 0.0143 |
| ≥ 11 times | 1.66 (1.16–2.38) | 0.0059 | 2.41 (1.61–3.58) | <0.0001 |
| **E19 Other students made sexual jokes, comments, or gestures that hurt my feelings** | | | | |
| 1 time | 1.20 (1.02–1.41) | 0.0276 | 1.26 (0.97–1.62) | 0.0725 |
| 3–5 times | 1.32 (1.02–1.70) | 0.0309 | 1.43 (1.01–1.99) | 0.0353 |
| ≥ 11 times | 1.17 (0.88–1.56) | 0.2567 | 1.49 (1.05–2.11) | 0.0224 |
| **E 20 I have received a threatening or hurtful message from another student in an email, on a website, on a cell phone, from text messaging, in an internet chat room, or in instant messaging** | | | | |
| 1 time | 1.28 (1.05–1.55) | 0.0111 | 1.51 (1.14–1.98) | 0.0031 |
| 3–5 times | 1.52 (1.09–2.12) | 0.0133 | 1.64 (1.10–2.42) | 0.0134 |
| ≥ 11 times | 1.49 (1.10–3.31) | 0.0513 | 2.18 (1.41–3.37) | 0.0004 |
| **E 21 Something hurtful has been shared about me on social media (Facebook, Twitter, Snapchat, etc.)** | | | | |
| 1 time | 1.03 (0.85–1.25) | 0.7074 | 1.05 (0.79–1.39) | 0.7029 |
| 3–5 times | 1.10 (0.77–1.58) | 0.5937 | 1.52 (0.99–2.31) | 0.0502 |

*(Continued)*

**Table 6.** (Continued)

| In the last 30 days, how many times have you been bullied at school in the ways listed below | During the past 12 months, did you ever feel so sad or hopeless every day for 2 weeks or more in a row that you stopped doing some usual activities? | | During the past 12 months, have tried to kill yourself? | |
|---|---|---|---|---|
| | Odds Ratio (95% CI) | P value | Odds Ratio (95% CI) | P value |
| ≥ 11 times | 0.83 (0.55–1.24) | 0.3716 | 2.26 (1.47–3.47) | 0.0002 |

95% Odds ratios that overlap 1 are not considered significant.

medication may influence emotions of sadness or hopelessness or a desire to commit suicide, we adjusted all ORs to reflect medication prescription and medication use. The results are shown in Table 5.

*Physical bullying.* When adjusting for psychotropic medication being prescribed or being taken, all aORs for physical bullying for both feeling sad or hopeless or attempting suicide overlapped 1 and were not significant.

These are similar to unadjusted ORs for physical bullying and feelings of hopelessness and are less than ORs for suicide. The aORs did not increase monotonically with frequency.

*Social bullying.* The aORs for social bullying varied according to the subtype of social bullying. All aORs for name calling for all forms of medication use and both sadness or hopelessness and suicide attempts overlapped 1 and were not significant while the unadjusted ORs were between 1.12 and 1.42. This is less than unadjusted ORs regarding both suicide attempts and feeling sad.

Social exclusion bullying had no significant association on suicide attempts, but aORs for 3–5 or more experiences of bullying ranged from 1.32 to 1.81 for sadness or hopelessness. The aORs were significant for both being prescribed and taking psychotropic medications.

Rumor mongering (E15) appeared to have the largest impact on both sadness or hopelessness and suicide attempts. With the exception of a single experience of rumormongering on suicide attempts (the aOR interval overlapped 1; the unadjusted OR was 1.37), the aORs for all experiences of bullying were significant and ranged from 1.41 to 2.05. The respective unadjusted ORs were between 1.41 and 2.15.

In all cases of social exclusion and rumor mongering, the aORs were also found to increase with increased experiences of bullying.

*Identity bullying.* The aORs for identity bullying also varied according to type. As with the unadjusted ORs for bullying based on religion, no significant association was demonstrated. Most 95% CIs overlapped or were below 1.

Racial bullying had no significant association on aORs for sadness or hopelessness for both those prescribed and those taking psychotropic medications. The 95% CIs either overlapped 1 or were below 1. This was not the case with the unadjusted ORs, which were significant for 1 and 3–5 episodes.

In 2 instances however, racial bullying increased the likelihood of suicide attempts. A single experience of racial bullying had an aOR of 1.44 to 1.54 for those prescribed and taking medication, respectively. Eleven or more experiences of being bullied had an aOR of 1.45 for those prescribed medication but those taking medication had 95% Cis that overlapped 1. The only significant unadjusted OR relating to suicide happened with 3–5 experiences, where the OR was 1.34.

Once again, bullying related to sexual orientation or gender identity or unwanted sexual comments had the biggest impact. All experiences of bullying about sexual orientation or gender orientation had aORs from 1.28 to 2.99. This was the case for those prescribed and those taking medicine as well as those experiencing sadness or hopelessness or attempting suicide. In all cases, aORs increased when greater experiences of bullying occurred. Hurtful sexual comments (E19) had lower aORs, ranging from 1.28 to 1.58 and aORs tended to increase the more times bullying was experienced. The aORs for identify did not always increase with the number of episodes of bullying.

*Cyberbullying.* In most cases, cyberbullying had no significant association with feelings of hopelessness or suicide ideation. The biggest exception was for hurtful messages on suicide attempts. With one exception, aORs for suicide attempts ranged from 1.35 to 1.99 for both those prescribed and taking medication. The unadjusted ORs were between 1.35 and 1.70.

Bullying on social media had significant associations for sadness or hopelessness for a single experience of bullying (aORs 1.46–1.67), but the aORs became non-significant with greater experience of bullying. These were higher than the unadjusted ORs which were clustered from 1.19 to 1.24

Significant associations of social media bullying on suicide were seen only for those who had experienced 11 or more episodes of bullying (aORs 1.99–2.03). All unadjusted ORs for suicide were significant ranging between 1.28 and 1.88 for 11 or more experiences.

## Bully-victims

As noted by others, students who experience bullying are sometimes bullies themselves [3, 8, 9]. Accordingly, we adjusted the risk for sadness or hopelessness and suicide attempts for students who replied that they had been a bully on 1 or more occasions. These results are shown above in Table 6.

*Physical bullying.* Physical bullying had little association except on suicide attempts for bullies who had been physically bullied on a single occasion. All other experiences of physical bullying either had 95% Cis that overlapped or were below 1.

*Social bullying.* The biggest associations of social bullying were the associations of social exclusion (aORs 1.24–1.95) or rumormongering (aORs 1.29–2.27) on feeling sad or hopeless. No form of social bullying had a significant association on suicide attempts.

*Identity bullying.* Non-sexual identity or gender orientation-based bullying had relatively few significant associations. Bully-victims who had a single experience of religious bullying had an aOR of 1.26 but no other associations of religious bullying were seen. Bully-victims who were racially bullied 3–5 times had an aOR of 1.85 but no other significant associations of racial bullying were seen.

As was seen above with the unadjusted ORs, sexual identity or gender orientation bullying, and hurtful sexual comments had the most impact among the various kinds of identity bullying. The biggest association was seen for sexual identity or gender orientation-based bullying (aOR 1.86–2.41) on bully-victims who attempted suicide. More modest associations of sexual identity or gender orientation bullying (aORs 1.59–1.66) were seen for bully-victims who experienced sadness or hopelessness. Hurtful sexual comments had the biggest association on suicide attempts by bully-victims who had been bullied 3 or more times (aORs 1.43–1.49).

*Cyberbullying.* Cyberbullying significantly increased aORs for both sadness or hopelessness and suicide attempts for all bully-victims who had experiences receiving a hurtful or threatening message (aORs 1.28–2.18). Cyberbullying on social media was less impactful. Significant aORs were seen only for ≥ 11 experiences of bully-victims being bullied who attempted suicide (aOR 2.26).

## Discussion

Our hope is that results such as ours may be helpful in the creation revision, and implementation of school anti-bullying policies and programs. The oldest and most widely cited is the program developed in Norway by Olweus [26–28]. The core components of the program included activities at the school level (school conference day, improved schoolyard supervision), the classroom level (class meetings and rules against bullying), and at the individual level (talking with bullies and victims, talking with parents). The program was found to have a marked effect to decrease bullying as well as other forms of antisocial behavior. The Olewus program has been adopted widely, including in the United States [29, 30]. and has since been updated to include information about cyberbullying and sexual or gender-based bullying. [31] Our results would support making these updates.

Other anti-bullying programs and attendant evaluations were reported by Evans and colleagues and Albayrak and co-workers [32, 33]. The Evans paper was a systematic review of 32 studies that examined 24 bullying interventions [32]. Overall, the results of these programs were mixed. Reviews of anti-bullying programs outside the United States, which involved fairly homogeneous student populations compared to the United States, reported more positive results than those focused on programs in the United States. In a study performed in Turkey, Albayrak and colleagues found that an anti-bullying program based on two theoretical models was effective in preventing bullying [33].

In a recent meta-analysis, Gaffney and colleagues reviewed 100 studies of anti-bullying programs [34]. In general, programs reduced bullying by 15–20%, but the authors noted that study populations were heterogeneous, as were study designs leading to differences in effect sizes.

Descriptions of anti-bullying programs in the literature generally do not provide much guidance as to the content that should be included when developing an such programs. Our literature review suggests areas, some of which are not well-discussed in the literature, where our results may have a useful influence as discussed below.

Although some literature suggests that the negative correlations from bullying increase positively with increased experiences of being bullied, this does not align with our results [13, 16]. Program design may thus not necessarily have to focus on the frequency of bullying.

Based on our results and our literature review, the highest priorities for designing anti-bullying interventions should be sexual and gender identity bullying as well as cyber bullying, which supports the work by Berlan [35], Norris [36] Cantone et al. discuss other possible interventions for cyber bulling [37]. Goodenow et al. suggest that interventions for gender identity may also have a positive effect [38].

In most cases, social bullying was positively correlated with both feeling sad or hopeless as well as suicide attempts which suggestions interventions should include specific information on name calling, social exclusion, and rumormongering.

While our results did not show much correlation between racial and religious bullying and mental health, it should be pointed out that Iowa is largely homogenous with respect to both race (White) and religion (Christian), and by no means are we suggesting that anti-bullying programs should not combat racial and religious bullying, especially in states or districts with a more heterogeneous student population than Iowa. We would also note that the data in this study were collected before the murder of George Floyd and the rise of the Black Lives Matter movement. Subsequent studies may find that racial bullying has stronger correlations than we did with the need for specific programming to be developed.

The lowest priority for program development appears to be in the area of physical bullying. Few positive correlations between physical bullying and mental health were seen. That said, we do not mean to imply that physical bullying should be ignored in program design.

Finally, the aORs for medication use and bully-victim status tended to be lower than unadjusted ORs. That said, it does not appear that anti-bullying programs narrowly targeted to students in these groups need to be developed. Unlike Brunstein and colleagues [16], we did not find that bully-victims were more prone to suicide than other student groups. As for unadjusted ORs, the aORS for bullying on sexual or gender identity had the strongest correlations with sadness or suicidality.

### Areas for future research

Research by Due and colleagues suggests that victims of bullying are more likely to take medication for pain or for psychological problems [18]. So, are the lower aORs we see a result of students taking psychotropic medicines who become less likely to be bullied or does the use of such pharmacotherapy mitigate against the effects of bullying? Or could the medication be a proxy for having a parent (and by extension a physician) take a student's complaints of psychological distress seriously, which in itself might lower the risk of sadness/hopelessness or suicide? While our methodology doesn't suggestion causation between psychotropic interventions and the mitigation of the effects of bullying, the fact that we see lower aORs for students taking psychotropic medicines is intriguing, and suggests that the role of such medicines in interventions should be studied in more detail.

Dake et al. found that teachers felt that drug use was of greater concern than bullying and that physical bullying was the worst type of bullying [39]. Work by the same researchers also found that the attitudes of principals had an effect on the acceptance and uptake of anti-bullying programs [40]. Over half of the principals surveyed were in the pre-contemplation phase of deciding to implement an anti-bullying program and 15% said that bullying was not a problem in their school. This suggests that the role of teachers and principals in anti-bullying programs be closely examined and perhaps rethought.

### Limitations

Given the number of independent variables we analyzed, we did not disaggregate our results by grade, sex, gender identity, race etc. It is unclear if our results apply equally across all cohorts of students in the IYS. Methodologically, this is a cross sectional study. Such methods can evaluate the correlation between variables but cannot be used to imply causation. We did not study the associations of poly-victimization in students who may have suffered (for example) sexual violence in addition to the types of bullying evaluated in the IYS. Our data does not allow us to determine if psychotropic medicines were started before or after a student experienced bullying. Nor do we know the type of medications respondents were prescribed. It should be noted, too, that the 2018 IYS did not specifically ask about gender identity.

As with any survey, social desirability bias may have limited respondents' willingness to provide truthful responses. Similarly, recall bias may have affected respondents' ability to accurately state their feelings or number of episodes of bullying in the past. These limitations notwithstanding, it is quite possible that with over 70,000 respondents, regression to the mean could reduce these potential biases.

### Conclusions

Perhaps the greatest take-home message from this study is that not all forms of bullying are equal in the harm they do. As educators or policy makers propose interventions to reduce bullying, it appears that interventions designed to address sexual identity or gender-based bullying are the most urgent.

Although there are differences between ORs and aORs, the data make it fairly clear that anti-bullying campaigns should target cyberbullying and identity-based bullying first, and more specifically sexual identity or gender-based joking. Despite the trend that aORs tend to return lower values, it is essential to observe that bullying problems in both the medicated and victim-bullies often resemble the unadjusted ORs. This suggests that educators need not create anti-bullying interventions directed exclusively towards students on medication or bully-victims. However, while degrees of suffering may vary somewhat between groups, there are also important similarities, so any victims of bullying should of course be included in interventions.

The role of medication in lessening the impact of bullying on victims is not clear. While our research suggests that medications may mitigate the effects of some types of bullying, we cannot draw any such conclusion from it. Finally, our results confirm that many bully-victims are also among those whose suffering is part of the dynamics of bullying.

## Author Contributions

**Conceptualization:** Daniel S. Alexander, John P. Rovers.

**Formal analysis:** Kaela L. Newman.

**Methodology:** Kaela L. Newman, John P. Rovers.

**Project administration:** Daniel S. Alexander, John P. Rovers.

**Supervision:** John P. Rovers.

**Writing – original draft:** Daniel S. Alexander, John P. Rovers.

**Writing – review & editing:** Kaela L. Newman, Daniel S. Alexander, John P. Rovers.

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
