## [Decision Letter · Decision Letter 0]

19 Jan 2022

PONE-D-21-34530Sadness, Hopelessness and Suicide Attempts in Bullying: Data from the 2018 Iowa Youth SurveyPLOS ONE

Dear Dr. Rovers,

Thank you for submitting your manuscript to PLOS ONE. After careful consideration, we feel that it has merit but does not fully meet PLOS ONE’s publication criteria as it currently stands. Therefore, we invite you to submit a revised version of the manuscript that addresses the points raised during the review process.

 The two reviewers addressed several major concerns about your manuscript. Please revise you manuscript carefully.

We look forward to receiving your revised manuscript.

Kind regards,

Kenji Hashimoto, PhD

Academic Editor

PLOS ONE

Journal Requirements:

Reviewers' comments:

Reviewer's Responses to Questions

**Comments to the Author**

1. Is the manuscript technically sound, and do the data support the conclusions?

Reviewer #1: Partly

Reviewer #2: Yes

2. Has the statistical analysis been performed appropriately and rigorously? 

Reviewer #1: Yes

Reviewer #2: Yes

3. Have the authors made all data underlying the findings in their manuscript fully available?

Reviewer #1: No

Reviewer #2: Yes

4. Is the manuscript presented in an intelligible fashion and written in standard English?

Reviewer #1: Yes

Reviewer #2: Yes

5. Review Comments to the Author

Reviewer #1: The study examined the effects of being involved in bullying either as a victim and/or a bully on mental health and suicide ideation by analyzing data from the 2018 Iowa Youth Survey.

The authors claim to have carried out an ecological study (see abstract), but the reasons that led the authors to this option are not clear enough. Analyzing the methodology and the results presented, it seemed to me to be a classic cross-sectional study. I do not observe cluster analysis or any type of aggregate that suggests that we are facing an ecological study.

Another important aspect is the reference (line 171) that 70,451 validated responses were available for analysis. What would be validated responses and what were the criteria for this validation? How many unvalidated responses were there in the dataset? What would the potential exclusion of unvalidated responses represent to the dataset?

Tables 4, 5 and 6 (pages 15-22) need to be better taken care of. Formatting is hampering the reading and interpretation of results, especially as tables take up many pages. It is also necessary to standardize the decimal places of the p-values. I see it as fundamental that the authors present descriptive statistics (n, %) for each variable tested

On the other hand, what stands out most in the manuscript was the strategy for discussing the results. The authors chose to debate subsets of data, segmenting the text and making it difficult to read and understand. This impaired the ability to compare the results with other research, making the interpretation superficial. Again, I emphasize that the study presents important results and a very relevant and urgent object. However, it deserves a thorough review of the text.

Finally, the authors do not make the data available for verification, merely informing that the data is the property of the Iowa Department of Public Health and can be made available upon request. I suggest that the form of data collection used in this manuscript be specifically informed.

Reviewer #2: Thank you for the opportunity to review this manuscript. Please find my comments below:

1. Abstract: The authors mentioned that the aim was to "...explore the effects of being involved in bullying...". As this was a cross-sectional study, cause-and-effect cannot be inferred. So I recommend to use words such as "associations" instead of "effects".

2. There is a typo on line 61, Pg 3: Cyberybullying

3. Would the authors consider reporting the values of the ORs and aORs in the abstract? It would provide information on the size of the association.

4. Introduction, Pg 4 Line 76: Please provide the context of Gan et al.'s study, since 55% is a comparatively high prevalence.

5. Information from the World Health Organization may be useful: https://www.who.int/news-room/fact-sheets/detail/youth-violence

6. Pg 4, ll. 82-83: Please provide the pg number for direct quotations

7. Pg. 5, ll. 105-109: A key reference on bullying and suicide attempt is:

Koyanagi, Ai, et al. "Bullying victimization and suicide attempt among adolescents aged 12–15 years from 48 countries." Journal of the American Academy of Child & Adolescent Psychiatry 58.9 (2019): 907-918.

Please consider citing it.

8. Pg. 6, l. 123 I think ADHD should be spelled attention-deficit/hyperactivity disorder.

9. Results: Pg. 30, Figure 1: X and Y axes could be reported in more detail

10. Results: Pg. 13, Table 1. In the table, you reported "In the last 30 days, how many times have you bullied someone else at school?" This is for bully behavior. However, I do not see the information for experiencing bullying. Is this information not available? If available, I think it would be important to report in this table. Also, the information of bully-victims would be important as well.

11. Discussion: I find a large part of the discussion merely reporting the results. A deeper engagement with the results could involve comparisons with other studies, I suggest that the discussion, esp the first 4 paragraphs of the discussion section be revised extensively.

12. I think that referring to the "chicken-and-egg problem" twice in the discussion may not be helpful in putting the study in context. Perhaps citing what other researchers have found, and how they explain the findings using past literature/theory could be more effective in providing context to your findings, i.e., linking the current study results with other extant findings in the US and the world?

13. Again, the discussion in Pg. 34-35 was only a repetition of the results, rather than a meaningful discussion of the findings. A major revamp of the discussion section is required.

6. PLOS authors have the option to publish the peer review history of their article (what does this mean?). If published, this will include your full peer review and any attached files.

Reviewer #1: No

Reviewer #2: No

---

## [Author Response · Author response to Decision Letter 0]

23 Jun 2022

Re: PONE-D-21-34530. Sadness, Hopelessness and Suicide Attempts in Bullying: Data from the 2018 Iowa Youth Survey

To the Editor:

We would like to express our thanks to yourself and both reviewers for the constructive comments concerning the above-mentioned paper. We have made all of the required changes possible and address each comment as noted below. We would re-iterate one concern, regarding availability of data. The dataset remains the property of the Iowa Department of Public Health and cannot be shared without permission. Information for how interested parties can inquire into accessing the data is included in our comments below.

Editor/Reviewer Comments and our responses follow:

Editor: Have the authors made all data underlying the findings in their manuscript fully available?

Response: We are contractually unable to comply with this request. Data for the IYS is collected by the State Departments of Education and Public Health. The data is cleaned and tabulated by the Iowa Consortium for Substance Abuse Research and Evaluation at the University of Iowa, and the data remains the property of the Department of Public Health and considered confidential under State of Iowa, Department of Public Health Disclosure of Confidential Public Health Information, Records, or Data Policy#CO 01-16-002. The publicly available report is available from https://iowayouthsurvey.idph.iowa.gov/. See also reference 18.

Reviewer #1: The authors claim to have carried out an ecological study (see abstract), but the reasons that led the authors to this option are not clear enough.

Response: Upon reflection, we agree with the reviewer and have clarified the abstract and body of the paper, calling this a cross-sectional study.

Reviewer #1: Another important aspect is the reference (line 171) that 70,451 validated responses were available for analysis. What would be validated responses and what were the criteria for this validation? How many unvalidated responses were there in the dataset? What would the potential exclusion of unvalidated responses represent to the dataset?

Response: The methods section has been expanded to describe more fully the data collection and cleaning process. The Department of Public Health provided a cleaned data set. We are unable to determine the criteria by which the Iowa Consortium for Substance Abuse Research and Evaluation at the University of Iowa cleaned the data or if exclusion of the unvalidated responses could have influenced the results or our conclusions. 

Reviewer #1: Tables 4, 5 and 6 (pages 15-22) need to be better taken care of. Formatting is hampering the reading and interpretation of results, especially as tables take up many pages. It is also necessary to standardize the decimal places of the p-values. I see it as fundamental that the authors present descriptive statistics (n, %) for each variable tested.

Response: The decimal places for all p-values in Tables 4,5, and 6 have all been standardized to 4 significant digits. Adding descriptive statistics to Tables 4,5, and 6 would have resulted in even more cluttering of the tables. Accordingly, Tables 1 and 2 have been expanded to include the total number of responses, the proportion of students answering each question and the proportion indicating “Yes” or “≥ 1 time”. 

Reviewer #1: On the other hand, what stands out most in the manuscript was the strategy for discussing the results. The authors chose to debate subsets of data, segmenting the text and making it difficult to read and understand. This impaired the ability to compare the results with other research, making the interpretation superficial. Again, I emphasize that the study presents important results and a very relevant and urgent object. However, it deserves a thorough review of the text.

Response: We take the reviewer’s point seriously and have essentially re-written the entire discussion to reflect interpretation of the data over simply reporting it.

Reviewer #1: Finally, the authors do not make the data available for verification, merely informing that the data is the property of the Iowa Department of Public Health and can be made available upon request. I suggest that the form of data collection used in this manuscript be specifically informed.

Response: As noted above we are contractually prohibited from making the data set available. Interested parties can contact the Iowa Department of Public Health at https://idph.iowa.gov/PublicHealthData/data-requests and request access to the data.

Reviewer #2: Abstract: The authors mentioned that the aim was to "...explore the effects of being involved in bullying...". As this was a cross-sectional study, cause-and-effect cannot be inferred. So I recommend to use words such as "associations" instead of "effects".

Response: The reviewer is correct. All references to “effects” have been replaced with “association(s)”.

Reviewer #2: There is a typo on line 61, Pg 3: Cyberybullying

Response: Thank you. This has been corrected.

Reviewer #2: Would the authors consider reporting the values of the ORs and aORs in the abstract? It would provide information on the size of the association.

Response: ORs have been added. aORs are more variable and likely to differ in significance more than ORs. We felt that reporting aORs in the abstract would be more confusing and have left the wording as “… generally lower than unadjusted ORs.”

Reviewer #2: Introduction, Pg 4 Line 76: Please provide the context of Gan et al.'s study, since 55% is a comparatively high prevalence. 

Response: We can only speculate as to why Gans et al. had such a high prevalence. We note that they used different instruments to evaluate bullying and that the most frequent forms of bullying in their study were name calling and teasing (40%). These results are similar to our results of social bullying in 35-39% of respondents. 

Reviewer #2: Information from the World Health Organization may be useful: https://www.who.int/news-room/fact-sheets/detail/youth-violence. 

Response: This has been added as Reference 11.

Reviewer #2: Pg 4, ll. 82-83: Please provide the pg number for direct quotations

Response: We are not sure how to respond to this. The quotation is from a journal article, not a book. Page numbers are not generally included for quotations from articles, to our understanding. 

Reviewer #2: Pg. 5, ll. 105-109: A key reference on bullying and suicide attempt is:

Koyanagi, Ai, et al. "Bullying victimization and suicide attempt among adolescents aged 12–15 years from 48 countries." Journal of the American Academy of Child & Adolescent Psychiatry 58.9 (2019): 907-918. Please consider citing it.

Response: This has been cited as reference 2.

Reviewer #2: Pg. 6, l. 123 I think ADHD should be spelled attention-deficit/hyperactivity disorder.

Response: The change has been made. 

Reviewer #2: Results: Pg. 30, Figure 1: X and Y axes could be reported in more detail

Response: Upon reflection, we did not feel that figure 1 added very much to the discussion, so it has been eliminated. The subsection headed “Variations in Unadjusted ORs” on page 34 provides a clearer explanation of the point we were trying to make. 

Results: Pg. 13, Table 1. In the table, you reported "In the last 30 days, how many times have you bullied someone else at school?" This is for bully behavior. However, I do not see the information for experiencing bullying. Is this information not available? If available, I think it would be important to report in this table. Also, the information of bully-victims would be important as well.

Response: Table 2 has been modified to include the proportion of respondents answering each bullying question and the proportion answering ≥ 1 time. There is no single table in the results that totals up a single number of the number of students who experienced bullying. 

Reviewer #2: Discussion: I find a large part of the discussion merely reporting the results. A deeper engagement with the results could involve comparisons with other studies, I suggest that the discussion, esp the first 4 paragraphs of the discussion section be revised extensively.

Response: This comment is similar to that of Reviewer 1. We agree and the discussion section has been largely re-written. 

Reviewer #2: I think that referring to the "chicken-and-egg problem" twice in the discussion may not be helpful in putting the study in context. Perhaps citing what other researchers have found, and how they explain the findings using past literature/theory could be more effective in providing context to your findings, i.e., linking the current study results with other extant findings in the US and the world?

Response: We agree with the reviewer. This is not essential to the paper and references to it have been deleted. 

Reviewer#2: Again, the discussion in Pg. 34-35 was only a repetition of the results, rather than a meaningful discussion of the findings. A major revamp of the discussion section is required.

Response: See previous responses regarding discussion.

We trust that we have been able to address all editorial and reviewers’ concerns adequately and look forward to receiving a decision on the final disposition of the manuscript.

Sincerely,

John Rovers

John Rovers, PharmD, MIPH

Professor of Pharmacy & Health Sciences

John.Rovers@drake.edu

c.c. Dan Alexander

 Kaela Newman

---

## [Decision Letter · Decision Letter 1]

12 Jul 2022

PONE-D-21-34530R1Sadness, Hopelessness and Suicide Attempts in Bullying: Data from the 2018 Iowa Youth SurveyPLOS ONE

Dear Dr. Rovers,

Thank you for submitting your manuscript to PLOS ONE. After careful consideration, we feel that it has merit but does not fully meet PLOS ONE’s publication criteria as it currently stands. Therefore, we invite you to submit a revised version of the manuscript that addresses the points raised during the review process.

The two reviewers addressed several minor concerns about your revised manuscript. Please revise your manuscript carefully.

We look forward to receiving your revised manuscript.

Kind regards,

Kenji Hashimoto, PhD

Section Editor

PLOS ONE

Journal Requirements:

Reviewers' comments:

Reviewer's Responses to Questions

**Comments to the Author**

1. If the authors have adequately addressed your comments raised in a previous round of review and you feel that this manuscript is now acceptable for publication, you may indicate that here to bypass the “Comments to the Author” section, enter your conflict of interest statement in the “Confidential to Editor” section, and submit your "Accept" recommendation.

Reviewer #1: All comments have been addressed

Reviewer #2: (No Response)

2. Is the manuscript technically sound, and do the data support the conclusions?

Reviewer #1: Yes

Reviewer #2: Partly

3. Has the statistical analysis been performed appropriately and rigorously? 

Reviewer #1: Yes

Reviewer #2: Yes

4. Have the authors made all data underlying the findings in their manuscript fully available?

Reviewer #1: No

Reviewer #2: No

5. Is the manuscript presented in an intelligible fashion and written in standard English?

Reviewer #1: Yes

Reviewer #2: Yes

6. Review Comments to the Author

Reviewer #1: I appreciated the way the authors handled my comments and suggestions. Examining the current version of the manuscript, I realize that my recommendations were followed. On the other hand, I suggest that the authors review the limitation section of the study. Note that while the authors accepted the suggestion to revise the study design, the “limitations” section indicates that the study design is ecological. Please to review it.

Reviewer #2: I would like to thank the authors for the revision made, and for extensively addressing the comments, including reanalysis of the data to yield aORs. Please find my comments as follows:

Abstract, Results: "...adjusted ORs (aORs) were generally lower than unadjusted ORs." This statement is too general to be meaningful. I would prefer if the authors state what the changes were, and if any of the predictors had become non-significant in the fully adjusted model.

Abstract, Methods: Is this an ecology study? If not please remove the statement "The study used an Ecological Study method".

Results, Table 3 and 4: Please standardise the reporting of p-values to 3 decimals.

Discussion: The discussion section needs to be revised extensively. A large part of it needs to be moved to the results section. Although the authors have made some effort to discuss the results in light of extant literature, there are some parts of the discussion which are still just repetitions of the results section. Perhaps the authors could read some examples of the discussion section of some high quality journal articles to understand how to synthesize their results and to discuss them meaningfully. Without an extensive revision of the discussion section, I'm afraid the manuscript would be unsuitable for publication.

Formatting: Quotations from a journal article should still include its page numbers. Please provide the page number(s) for the following quotation: "“A student is being bullied or victimized when he or she is 83 exposed repeatedly and over time to negative actions on the part of one or more students.” (2)". Reference#2 is a journal article. It should have a page number.

7. PLOS authors have the option to publish the peer review history of their article (what does this mean?). If published, this will include your full peer review and any attached files.

Reviewer #1: No

Reviewer #2: No

---

## [Author Response · Author response to Decision Letter 1]

19 Sep 2022

Reviewer #1: 

I appreciated the way the authors handled my comments and suggestions. Examining the current version of the manuscript, I realize that my recommendations were followed. On the other hand, I suggest that the authors review the limitation section of the study. Note that while the authors accepted the suggestion to revise the study design, the “limitations” section indicates that the study design is ecological. Please to review it.

Response: The wording has been changed to reflect that this was a cross-sectional survey.

Reviewer #2: 

Abstract, Results: "...adjusted ORs (aORs) were generally lower than unadjusted ORs." This statement is too general to be meaningful. I would prefer if the authors state what the changes were, and if any of the predictors had become non-significant in the fully adjusted model.

Response: This concern has been largely addressed in the body of the paper. The discussion section has been modified extensively and comparisons between ORs and aORs are made throughout.

Abstract, Methods: Is this an ecology study? If not please remove the statement "The study used an Ecological Study method".

The change has been made to reflect a cross-sectional method.

Results, Table 3 and 4: Please standardise the reporting of p-values to 3 decimals.

Response: We have left the data at 4 decimals. We would argue that the differences between 3 and 4 decimals is close to meaningless and accordingly, did not make any changes.

Discussion: The discussion section needs to be revised extensively. 

Response: The discussion section and the conclusions have been largely revised. We would acknowledge that the revised discussion continues to have some data in it (although less than before). We did so in order to make comparisons between unadjusted and adjusted odds ratios easier for the reader to see.

Formatting: Quotations from a journal article should still include its page numbers.

Response: We re-set the citation manager to create a proper reference list. Page numbers are included when appropriate.

---

## [Decision Letter · Decision Letter 2]

7 Oct 2022

PONE-D-21-34530R2Sadness, Hopelessness and Suicide Attempts in Bullying: Data from the 2018 Iowa Youth Survey

PLOS ONE

Dear Dr. Rovers,

Thank you for submitting your manuscript to PLOS ONE. After careful consideration, we feel that it has merit but does not fully meet PLOS ONE’s publication criteria as it currently stands. Therefore, we invite you to submit a revised version of the manuscript that addresses the points raised during the review process.

The reviewer #2 addressed several minor concerns about your manuscript again. Please revise your manuscript according to comments from the reviewer #2.

We look forward to receiving your revised manuscript.

Kind regards,

Kenji Hashimoto, PhD

Section Editor

PLOS ONE

Journal Requirements:

Reviewers' comments:

Reviewer's Responses to Questions

**Comments to the Author**

1. If the authors have adequately addressed your comments raised in a previous round of review and you feel that this manuscript is now acceptable for publication, you may indicate that here to bypass the “Comments to the Author” section, enter your conflict of interest statement in the “Confidential to Editor” section, and submit your "Accept" recommendation.

Reviewer #1: All comments have been addressed

Reviewer #2: (No Response)

2. Is the manuscript technically sound, and do the data support the conclusions?

Reviewer #1: Yes

Reviewer #2: Partly

3. Has the statistical analysis been performed appropriately and rigorously? 

Reviewer #1: Yes

Reviewer #2: Yes

4. Have the authors made all data underlying the findings in their manuscript fully available?

Reviewer #1: No

Reviewer #2: No

5. Is the manuscript presented in an intelligible fashion and written in standard English?

Reviewer #1: Yes

Reviewer #2: Yes

6. Review Comments to the Author

Reviewer #1: I appreciate the way the authors treat my suggestions. All the changes requested in the last opinion were fully complied with by the authors. Thus, there are no new suggestions to be made.

Reviewer #2: Thank you for the opportunity to review this manuscript again, which is much improved, and I commend the authors for their effort. This is an important work with very interesting findings to inform policy-making. While there are many merits, I'm afraid, however, that the discussion section is still below the par of a standard discussion for publication in an academic journal. For example, the authors are still reporting the results very extensively in the discussion section, as follows: "Other forms of identity bullying on race or religion also had broadly similar ORs across all categories of mental distress and medication use. (Significant race ORs 1.23-1.34. Religion ORs all non-significant. Significant race aORs 1.44-1.54. Religion aORs all non-significant.) Overall, the results in aORs support those unadjusted ORs in that sexual identity bullying should take the highest priority in developing anti-bullying programs. Those prescribed or taking medication had slightly lower but still significant aORS. Similar to identity bullying, aORs for social bullying were generally a bit lower than ORs. (ORs 1.23 – 2.15. aORs Not significant for name calling, otherwise 1.32-1.14 for sadness only, and 1.41 – 2.05 for suicide attempts)"

This is not followed by a meaningful discussion of the results. Please delete this part of the discussion, along with other extensive reporting of the results in the discussion section. I feel that this kind of reporting violates a very basic notion of separating the results from the discussion section. Repeating the results in the discussion does not make it a meaningful discussion of their findings. An oversimplified way of explaining the difference between the results and discussion section is that the results reports "what" and the discussion discusses "how, why, what is important for the future" etc.

There are also a few phrases/sentences in page 38: "Significant race ORs 1.23-1.34. Religion ORs all non-significant. Significant race aORs 1.44-1.54. Religion aORs all non-significant.)" This standard of writing is not acceptable in an academic journal. These are phrases that are incomplete sentences. I suggest the authors revise their way of writing throughout. Also, please standardise the decimals to 4 or 5 decimals. This is another important part of standard reporting in an academic journal. There are some basic rules, no matter how minor, when writing a manuscript for publication.

There is a very short paragraph, "No aOR for physical bullying had a significant association with impaired mental health for

students prescribed or taking psychotropic medicines". Please combine this paragraph with the next paragraph in which you discussed the implications of the findings. I would recommend that the manuscript be edited by an editor who is familiar with academic writing to improve the way the discussion section is presented. The authors could also read other articles from high impact journals for examples of a well-written discussion section. The authors will notice that they almost never have this kind of extensive reporting of their results in the discussion.

7. PLOS authors have the option to publish the peer review history of their article (what does this mean?). If published, this will include your full peer review and any attached files.

Reviewer #1: **Yes: **Ricardo de Mattos Russo Rafael

Reviewer #2: No

---

## [Author Response · Author response to Decision Letter 2]

11 Jan 2023

We would like to express our thanks to yourself and both reviewers for the constructive comments concerning the above-mentioned paper. We have made significant changes in the paper and are re-submitting a revised manuscript.

We note that the Editor and Reviewer #1 had no further recommended changes.

We also note that Reviewer #2 continued to have major concerns with our Discussion Section. Most of these concerns focus on presentation of Results in the Discussion and they point out a number of areas of confusion.

Rather than try and address these suggestions one by one, we have completely re-written the Discussion Section, which we would argue addresses these concerns wholistically. At the risk of sounding unserious, we eventually came to the conclusion that what Reviewer #2 was trying to say was, “Those results are nice, but so what?” Accordingly, our significantly revised Discussion attempts to place our results within the context of previously published papers on various anti-bullying programs. Our review of previous work in the discussion is then merged with what our results imply. That is: What content should be included in anti-bullying programs and which kinds of bullying are the highest priority? We also include a section on possible areas for future research.

We hope that this major revision is adequately addresses Reviewer #2’s concerns and that our paper can now be accepted for publication.

---

## [Editor Report · Decision Letter 3]

16 Jan 2023

Sadness, Hopelessness and Suicide Attempts in Bullying: Data from the 2018 Iowa Youth Survey

PONE-D-21-34530R3

Dear Dr. Rovers,

We’re pleased to inform you that your manuscript has been judged scientifically suitable for publication and will be formally accepted for publication once it meets all outstanding technical requirements.

Kind regards,

Kenji Hashimoto, PhD

Section Editor

PLOS ONE
---

## [Editor Report · Acceptance letter]

20 Jan 2023

PONE-D-21-34530R3 

Sadness, Hopelessness and Suicide Attempts in Bullying: Data from the 2018 Iowa Youth Survey 

Dear Dr. Rovers:

I'm pleased to inform you that your manuscript has been deemed suitable for publication in PLOS ONE. Congratulations! Your manuscript is now with our production department. 

Kind regards, 

on behalf of

Prof. Kenji Hashimoto 

Section Editor

PLOS ONE